

# Composite environmental indices—a case of rickety rankings

Shelley M. Stevens[1], Michael K. Joy[2], Wokje Abrahamse[1], Taciano L. Milfont[3] and Lynda M. Petherick[1,†]

[1] School of Geography, Environment and Earth Sciences, Te Herenga Waka Victoria University of Wellington, Wellington, New Zealand
[2] School of Government, Te Herenga Waka Victoria University of Wellington, Wellington, New Zealand
[3] School of Psychology, Te Whare Wananga o Waikato University of Waikato, Tauranga, New Zealand
[†] Deceased.

## ABSTRACT

Composite indices have been widely used to rank the environmental performance of nations. Such environmental indices can be useful in communicating complex information as a single value and have the potential to generate political and media awareness of environmental issues. However, poorly constructed, or poorly communicated indices, can hinder efforts to identify environmental failings, and there are considerable differences in rank among existing environmental indices. Here, we provide a review of the conceptual frameworks and methodological choices used for existing environmental indices to enhance our understanding of their accuracy and applicability. In the present study, we review existing global indices according to their conceptual framework (objectives of the index and set of indicators included) and methodological choices made in their construction (*e.g.*, weighting and aggregation). We examine how differences in conceptual frameworks and methodology may yield a more, or less, optimistic view of a country's environment. Our results indicate that (1) multidimensional environmental indices with indicators related to human health and welfare or policy are positively correlated; (2) environment-only indices are positively correlated with one another or are not correlated at all; (3) multidimensional indices and environment-only indices are negatively correlated with each other or are not correlated at all. This indicates that the conceptual frameworks and indicators included may influence a country's rank among different environmental indices. Our results highlight that, when choosing an existing environmental index—or developing a new one—it is important to assess whether the conceptual framework (and associated indicators) and methodological choices are appropriate for the phenomenon being measured and reported on. This is important because the inclusion of confounding indicators in environmental indices may provide a misleading view of the quality of a country's environment.

Corresponding author
Shelley M. Stevens,
shelley.stevens@vuw.ac.nz

# INTRODUCTION

Composite environmental indices attempt to communicate complex information by combining multiple indicators into a single score. They provide an overall snapshot of

some features of environmental systems and allow for comparisons between environmental systems (*Oțoiu & Grădinaru, 2018*). Environmental indices can be useful tools to measure stability and change over time (improvement or decline), easily communicate information to the general public and policymakers, drive accountability between countries, encourage accountability of a nation's government to its citizens, and facilitate public engagement (*Dobbie & Dail, 2012*; *Nardo et al., 2005*). Environmental indices can also help inform and support policy decisions, as well as generate political and media awareness of environmental issues (*Haberland, 2008*). However, if indices are poorly constructed or communicated, they can hinder efforts to identify and address environmental failings and may mislead policy messages and decisions (*Alberti & Parker, 1991*; *Haberland, 2008*).

The popularity of environmental indices has grown in recent years. Environmental indices have been used to measure and compare results for individual nations in global analyses (*Bradshaw, Giam & Sodhi, 2010*; *Wackernagel, Beyers & Rout, 2019*; *Wendling, Emerson & Esty, 2020*), for clusters of nations (*Cook et al., 2017*; *Halkos & Zisiadou, 2018*), for regions within a nation (*Mukherjee & Kathuria, 2006*; *Universiti Teknologi, 2018*), and to measure the state of specific ecosystems (*Blumetto et al., 2019*). Most commonly, however, environmental indices are used in global studies to compare results across nations, because national policy formulation and implementation are centred at the national level (*Roberts, 2011*; *Usubiaga-Liaño & Ekins, 2021*). Therefore, in this review, the geographic scale is for individual nations within global analyses.

There is a diverse range of available environmental indices, but country rankings differ considerably in these different indices. To illustrate, national rankings vary by an average of 45 places between the 2018 versions of the Environmental Performance Index (EPI) and ecological footprint (EF) (ranks normalized 0–100, File S1). Such large rank variations, particularly for low income and high income countries, have led to questions about the legitimacy and validity of rankings (*Morse & Fraser, 2005*; *Press Information Bureau Government of India, 2022*). Some environmental indices include indicators such as access to clean drinking water and sanitation, mortality, and per capita income for which high income nations will score well and low income countries will score poorly. The inclusion of such indicators may provide a more or less optimistic view of the environment of a nation, depending on its development status (*e.g.*, Global North, Global South) (*Bradshaw, Giam & Sodhi, 2010*; *Haberland, 2008*).

There is extensive literature examining metrics for sustainable development and environmental sustainability (*Alberti & Parker, 1991*; *Böhringer & Jochem, 2007*; *Bradshaw, Giam & Sodhi, 2010*; *Dobbie & Dail, 2012*; *Ebert & Welsch, 2004*; *Kwatra, Kumar & Sharma, 2020*; *Mallett, 1999*; *Oțoiu & Grădinaru, 2018*; *Siche et al., 2008*). However, past work has largely focused on the need for credible environmental metrics (*Alberti & Parker, 1991*; *Hák, Janoušková & Moldan, 2016*), principles for constructing composite indices (*Dobbie & Dail, 2012*; *Floridi et al., 2011*; *Nardo et al., 2005*), and weighting, aggregation, and uncertainty analysis methods (*Burgass et al., 2017*; *Gan et al., 2017*; *Greco et al., 2019*; *Morse & Fraser, 2005*; *Tripathi & Singal, 2019*). Research into rank differences between existing environmental indices is relatively sparse. Research into rank variations in other fields, such as university rankings and sports leagues, is also limited and has largely

concentrated on construction methods and policy implications (*Garcia-Zorita et al., 2018*; *Saisana, d'Hombres & Saltelli, 2011*).

The aim of this review is to answer the following questions: Which conceptual framework(s) or indicators inform the development of the country rankings? What effect do different conceptual frameworks (and indicators included have on global rankings? To what extent do the different environmental indices adequately capture environmental issues in low income and high income countries? In doing so, this review provides insights into how environmental indices are constructed and provide recommendations for best practices. In the present study, we have identified ten environmental indices for which country rankings can be compared to improve our knowledge of how conceptual frameworks and methodological choices (*e.g.*, weighting and aggregation methods) may influence rank variation. This article provides a novel contribution to the development and use of composite environmental indices. This review is intended for environmental scientists interested in environmental assessment and protection, conservation executors, environmental analysts, and policymakers at both the national and international levels.

The article is organized as follows. First, we discuss how different understandings of sustainable development, environmental sustainability, and natural capital have informed environmental indices. Second, we discuss differences in the conceptual frameworks and methodological choices used to develop environmental indices. Third, we summarize existing environmental indices and the conceptual frameworks and construction choices behind them. Fourth, we examine correlations between ranks in global environmental indices. Fifth, we examine ranks for two high income countries (Germany and New Zealand) and two low income nations (Mongolia and Niger) and identify potential reasons for wide differences in rank for low and high income nations generally. Finally, we discuss our key findings and outline how conceptual framework and methodological choices may affect the ranking of nations in environmental indices.

## REVIEW OF GLOBAL ENVIRONMENTAL INDICES

The Brundtland Commission (*Brundtland, 1987*) defined sustainable development as 'development that meets the needs of the present without compromising the ability of future generations to meet their own needs'. The three pillars (or dimensions) of sustainability are economy, environment, and society (see Fig. 1). The aim of sustainable development is to use economic development to promote fairer societies within the ecological carrying capacity of the planet (*Baker, 2016*; *Strange & Bayley, 2008*).

There are complex connections and trade-offs between economic growth, improvements in social state, and environmental quality (*Strange & Bayley, 2008*). The Environmental Kuznets Curve (EKC) hypothesises that environmental degradation initially increases with income, then declines (*Stern, Common & Barbier, 1996*). However, critics have highlighted that the EKC model is debateable, particularly when considering different indicators of environmental degradation. For example, biodiversity may not improve with rising income (*Mills & Waite, 2009*). Several studies have demonstrated that economic growth has a strong positive effect on carbon dioxide, sulphur dioxide, and industrial greenhouse
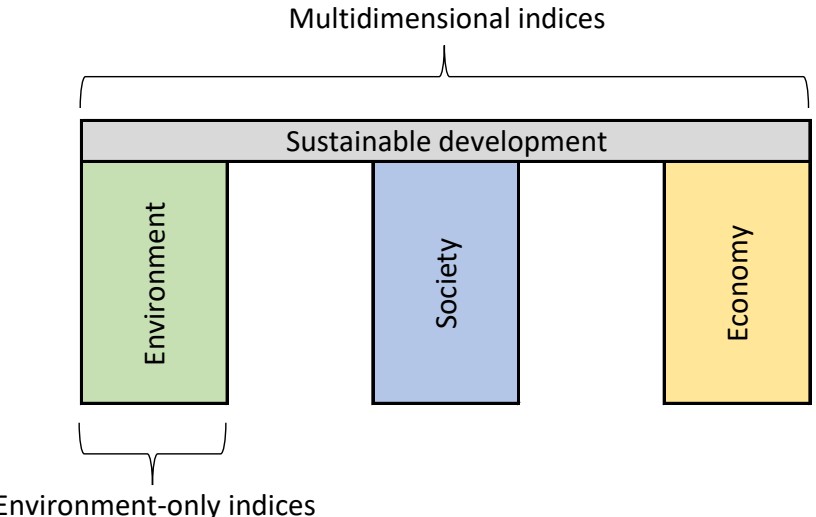

**Figure 1** **Diagram depicting the three dimensions of sustainability, adapted from Purvis et al. (2018).**
Note: Multidimensional indices include indicators that fall under the environment, society, and economy
pillars. Environment-only indices include indicators that only fall under the environment pillar.

gas (GHG) emissions, but weaker effects on particulate matter (PM) and non-industrial
GHG emissions (*Holtz-Eakin & Seldon, 1995*; *Stern, 2017*). The constructors of the pENV
investigated relationships between per capita wealth and environmental impact and found
that EKC predictions were not supported (*Bradshaw, Giam & Sodhi, 2010*). The EPI team,
however, found a positive correlation between EPI scores and country wealth—indicating
that rising GDP per capita is associated with higher EPI scores (*Wolf et al., 2022a*; *Wolf et
al., 2022b*).

While complex interactions exist between the three pillars of sustainability,
environmental sustainability can be distinguished from social sustainability and economic
sustainability (*Goodland, 1995*; *Usubiaga-Liaño & Ekins, 2021*). Several different definitions
for environmental sustainability exist (Table 1). A common theme in these definitions is
the need for some qualities or functions of natural capital to be sustained for future
generations. Natural capital (the natural environment) is comprised of natural resource
stocks (*e.g.*, water, forests, soil, wetlands, atmosphere), which provide a flow of goods
and services. Goods and services can be renewable and non-renewable with and without
marketed value (*Ekins et al., 2003*; *Helm, 2015*; *OECD, 2008*). Indicators representing the
qualities or functions of natural capital can be linked to environmental pressures, states,
and impacts, as well as social states where their functions are associated with human health
and welfare (*Ekins et al., 2003*; *Usubiaga-Liaño & Ekins, 2021*).

Environmental metrics represent the environmental dimension of sustainable
development and use a variety of indicators related to the qualities or aspects of natural
capital (*Usubiaga-Liaño & Ekins, 2021*). For the purposes of this review, we categorize
environmental indices into two broad categories. Environment-only indices focus
exclusively on aspects of natural capital that represent pressures, states, or impacts and

**Table 1  Definitions of environmental sustainability, adapted from *Usubiaga-Liaño & Ekins (2021)*.**

| Source | Definition |
|---|---|
| *Goodland (1995)* | Maintenance of natural capital. |
| *Holdren, Daily & Ehrlich (1995)* | Maintenance or improvement of earth's life support system. |
| *Ekins et al. (2003)* | Maintenance of important environmental functions, and therefore, the maintenance of the capacity of natural capital stocks to provide these functions. |
| *Sutton (2004)* | The ability to maintain qualities that are valued in the physical environment. |
| *Moldan, Janoušková & Hák (2012)* | Maintaining nature's services at a suitable level. |

exclude indicators related to environmental policies or human health and welfare. Thus, environment-only indices only include indicators that fall under the environmental pillar of sustainable development (Fig. 1). In line with research by *Bradshaw, Giam & Sodhi (2010)*, multidimensional indices use indicators that represent aspects of natural capital linked to environmental pressures, states, impacts as well as indicators related to environmental policies or human health and welfare. Human health and welfare functions provide services to humans which maintain health and contribute to human wellbeing in other ways—both economic and noneconomic (*Usubiaga-Liaño & Ekins, 2021*). Multidimensional indices typically incorporate (either directly or as calculation components) indicators such as human access to safe sanitation and drinking water, exposure to pollutants, and per capita income. Unlike environment-only indices, multidimensional indices include indicators that fall under all three pillars of sustainable development (Fig. 1).

## Considerations for constructing environmental indices

Composite indices are typically developed using the same basic steps, including development of a conceptual framework, data selection, imputation of missing data, weighting and aggregation, uncertainty and sensitivity analysis (Table 2). Some indices, however, may not use all the steps, or the conceptual framework and methodological choices may not be adequately described. This is concerning because results are often presented in the media without sufficient information regarding the conceptual framework or methods used to construct a given index (*Conrad & Cassar, 2018*). Additionally, misleading rankings can be misused by politicians and industry organizations (*Hsu, 2013*; *NewZealand Parliament, 2011*; *Rowarth, 2020*; *Rowarth, 2021*). For example, the director of two industry organizations representing dairy farmers in New Zealand has used the country's high ranking in the Legatum Prosperity Index (LPI) to argue that New Zealand's agricultural industry has a lower environmental impact than other countries (*Rowarth, 2021*), despite decades of scientific research suggesting otherwise  (*Duncan, 2014*; *Howard-Williams et al., 2010*; *Joy, 2015*). Thus, it is worthwhile to investigate how the conceptual frameworks and construction choices used in existing indices may influence rankings. In the case of the Natural Environment Pillar of the LPI (hereafter referred to as LPI-NE), the aim is to measure aspects of the physical environment that have a direct effect on people. Nevertheless, this index includes indicators that do not measure the physical environment, such as disability-adjusted life year (DALY) as a measure of disease burden and survey

**Table 2  Standard principles for constructing composite indices, adapted from *Dobbie & Dail (2012)* and *Nardo et al. (2005)*.**

| Step & reasoning |
| --- |
| 1. *Establishment of a conceptual framework*<br>- Provides criteria for the selection and combination of indicators under a fitness-for-purpose principle. |
| 2. *Data selection*<br>- Based on analytical soundness, measurability, regional coverage, and relevance of the indicators to the phenomenon being measured and their relationship to each other.<br>-  Quality of indicators should be checked. |
| 3. *Imputation of missing data*<br>- Required to provide a complete data set (*e.g.*, by means of single or multiple imputations).<br>- Missing variables may be estimated. |
| 4. *Multivariate analysis*<br>- Used to study the overall structure of the dataset, assess the suitability, and guide subsequent methodological choices (*e.g.*, weighting and aggregation). |
| 5. *Normalisation*<br>- Carried out to render the variables comparable. |
| 6. *Weighting and Aggregation*<br>- Performed to select appropriate weighting and aggregation procedures.<br>- Compensability among indicators should be allowed. |
| 7. *Uncertainty and sensitivity analysis*<br>- A multi-modeling approach may be used to build the index, and alternative conceptual scenarios for the selection of underlying indicators considered. |
| 8. *Return to the data*<br>- Required to reveal the main drivers for an overall good or bad performance.<br>- Transparency for good analysis and policymaking. |
| 9. *Links to other indicators*<br>- To correlate the composite indicator (or its dimensions) with existing (simple or composite) indicators as well as to identify linkages through regressions |
| 10. *Visualization of results*<br>- Visualization can influence or help to enhance interpretability. |

responses (*e.g.*, are you satisfied or dissatisfied with the quality of water?). This example illustrates the importance of choosing indicators that are consistent with the environmental phenomenon it aims to measure.

Several issues should be considered when assessing the adequacy of an environmental index and whether it is fit for purpose (*Bradshaw, Giam & Sodhi, 2010*; *Colling & Flynn, 2015*). Firstly, environmental indices differ according to the conceptual frameworks that inform them. Some environmental indices such as the EPI and Environmental State and Sustainability Index (ESSI) aim to measure how close nations are to established environmental policy targets (*Oțoiu & Grădinaru, 2018*; *Wolf et al., 2022a*; *Wolf et al., 2022b*), while others focus on the supply and demand of nature (the Ecological Footprint) (*Wackernagel, Beyers & Rout, 2019*), the health of ecosystems (Environmental Wellbeing Index (EWI) *Prescott-Allen, 2001*), or environmental impacts (absolute ENV (aENV) and proportional ENV (pENV)) (*Bradshaw, Giam & Sodhi, 2010*). A conceptual framework

should define the objectives of the index and provide a clear outline of what is being assessed (*i.e.,* what is the environmental phenomenon being measured?) (*Dobbie & Dail, 2012*; *Nardo et al., 2005*). However, if the conceptual framework for a given index lacks clear objectives and definitions of terms, and if selection criteria for indicators do not reflect the conceptual framework, this can lead to debate and confusion (*Haberland, 2008*; *Ihobe, 2013*).

Secondly, indicators should be relevant to the phenomenon being measured and data selection processes should be carefully considered to avoid introducing inappropriate or poor-quality indicators (or excluding key indicators). It is worth noting that there is often a compromise between scientific precision and information available—what scientists would like to measure does not always correspond with what can practically be measured at a given point in time (*Nardo et al., 2005*). However, if the available data sets are inadequate it can lead to questions about whether national ranks are meaningful (*Browning, 2011*; *Haberland, 2008*; *Ihobe, 2013*; *Press Information Bureau Government of India, 2022*). This was the case for the water quality index included in the 2010 EPI (*Emerson et al., 2010*). In New Zealand, freshwater scientists revealed flaws in the water quality index used and challenged the prime minister at the time about his misuse of New Zealand's rank for freshwater in the EPI (*Browning, 2011*; *NewZealand Parliament, 2011*). Subsequently the EPI team reviewed the indicator and it was removed from later versions of the index (*Hsu, 2013*). Another example of issues with data quality and coverage is the inclusion of renewable water resources and freshwater withdrawal in the LPI-NE (*Legatum Institute, 2020*). Because data is limited for large parts of the world, modelled data from the Food and Agriculture Organization (FAO) AQUASTAT database is used instead. The FAO itself raised the need for caution when using the dataset: "the problems of insufficient data and dubious accuracy paint a picture that, if interpreted as a final product, lead to incorrect assumptions" (*FAO, 2021*, p.41, para. 5). The FAO highlighted issues with data coverage for North America, Europe, Japan, Australia, and New Zealand—all of which rank highly in the LPI-NE (*FAO, 2021*). Failure to assess the impact of imputed values can impact on the quality of an index.

Another issue to consider is that data from different time periods may be used by the developers of environmental indices. A given index may include data for a particular point in time or averages for a set number of years. For example, the aENV and pENV use the most recent three-year average for annual fertilizer consumption and the most recent ten-year average for $CO_2$ emissions (*Bradshaw, Giam & Sodhi, 2010*). Others may include data for a particular point in time, trends over a certain time period, or future projections. For example, in the EPI forest loss is measured over a five-year period and compared to a reference year, GHG emissions are predicted for the year 2050, and human exposure to heavy metals is calculated for a 30-year period (*Wolf et al., 2022b*; *Wolf et al., 2022a*). While the appropriateness of timeframes may vary according to the indicator, it is crucial to clearly communicate the periods used so that potential users of the index can evaluate the decisions made by the developers. However, timeframes for data collection and decision-making are not clearly communicated for some indices. For example, timeframes

for the dataset used for the CIEP are not provided (*García-Sánchez, Almeida & Camara, 2015*), which adds another level of complexity when interpreting the index.

Finally, there are issues with the statistical methods used for some environmental indices. Different procedures may be used for normalization, weighting, aggregation, and uncertainty and sensitivity analysis. Weighting techniques are particularly important as they can lead to different rankings (*Chakrabartty, 2018*; *Dobbie & Dail, 2012*; *Morse & Fraser, 2005*). There are three common methods for weighting indicators: (1) expert opinion-based weighting, which uses subjective judgements in an open discussion with experts (*e.g.*, EPI), (2) equal weighting where all indicators are given the same weighting (*e.g.*, Environmental Vulnerability Index (EVI)), and (3) statistics-based weighting (*e.g.*, ESSI, aENV and pENV). Indices that use expert opinion-based and equal weighting methods have been criticized for failing to consider the interlinkages and dynamic interrelations of the various components (*Agliardi, Pinar & Stengos, 2014*; *Morse & Fraser, 2005*; *Paulvannan Kanmani et al., 2020*). Indices such as the aENV, pENV and ESSI use multivariate analysis to explore the nature of datasets and determine weights (*Bradshaw, Giam & Sodhi, 2010*; *Oțoiu & Grădinaru, 2018*). A recent study used stochastic dominance theory to investigate how sensitive EPI scores are to alternative weight choices (*Pinar, 2022*). By using alternative weights, 67 countries would have seen a rank change of at least 30 places, while 37 would have seen a rank change of at least 50 places. This example illustrates the importance of incorporating uncertainty and sensitivity analyses into the development of environmental indices, as well as communicating the statistical methods employed explicitly and transparently. It is important to highlight that statistical methods also have a drawback in that they cannot be easily communicated, which may reduce public trust in the index.

In summary, environmental indices vary widely in terms of conceptual framework (what environmental phenomena is being measured, terminology used, selection criteria for indicators) and methodological choices (data choices, normalization, weighting and aggregation techniques, uncertainty and sensitivity analysis). The quality of the data included in different indices also varies (*e.g.*, gaps in temporal and spatial coverage). The following section presents a brief characterization of ten global environmental indices.

## Summary of ten national-level indices
### Search method
An exhaustive search was performed to identify potential worldwide environmental indices (conducted by the first author). Two public search engines (Google and Bing) were used as well as academic databases (Scopus, JSTOR, Science Direct). The following search strategy was used: (Environmental AND (index OR indices OR composite)) AND (nation OR national OR countries OR performance OR quality OR health). Our characterization is restricted to ten national level indices that have sufficient published information to determine their properties. Of the ten indices, only the EPI, LPI-NE, Efpc and EFtot are ongoing projects and have recent versions available. We have included the remaining six indices as they provide valuable information on the strengths and weaknesses of environmental indices. Indices were considered in our analysis if they included at least 100 countries, if the weighting and aggregation methods were clear, and if there was

sufficient information about the indicators included (*e.g.*, data sources were transparent, and indicators clearly defined). To avoid researcher bias, the first and second authors evaluated each of the ten potential national-level environmental indices separately.

Information for each index is summarised in overview tables, including (1) a brief characterization of each index, (2) what each index aims to measure—and whether this it does so adequately or not, (3) the methods used to construct each index (according to the steps outlined in Table 2), and (4) their strengths and weaknesses.

Four indices (the CIEP, EPI, ESSI, and LPI-NE) are multidimensional indices, while the remaining indices (EFpc, EFtot, EVI, EWI, aENV, pENV) are environment-only indices. The weighting and aggregation methods used for multidimensional and environment-only indices are mixed (Tables 3 and 4). To illustrate, the four multidimensional indices use three different methods. The CIEP and ESSI use statistics-based weighting, the EPI uses expert opinion-based weighting, and the LPI-NE uses a combination of two methods where weights are primarily determined by expert opinion and academic literature, and to a lesser extent by statistical significance to the economic and social wellbeing of a country. Environment-only indices also use a mix of different methods. The aENV and pENV use statistics-based weighting, the EFpc and EFtot use expert opinion-based weighting, and the EVI and EWI use equal weighting.

## RANK CORRELATIONS IN GLOBAL ENVIRONMENTAL INDICES

There are considerable differences in rank among existing environmental indices. Here, we compare country ranks for each of the indices identified in 'Summary of ten national-level indices', and explore potential explanations for differences in ranking, such as conceptual framework and construction choices.

### Methods

First, we examined rank correlations for global environmental indices. Data for the EPI (2020), LPI-NE (2021), EF per capita (EFpc) and EF total (EFtot) (2018) were sourced from official websites (*Global Footprint Network, 2022*; *Legatum Institute, 2021*; *Yale Center for Environmental Law and Policy, 2023*). Data for ESSI (2018), EVI (1999), aENV and pENV (2010) were obtained from academic publications (*Bradshaw, Giam & Sodhi, 2010*; *Kaly, Pratt & Mitchell, 2004*; *Oṭoiu & Grădinaru, 2018*). Data for the CIEP (2015) was obtained directly from Dr Thiago Almeida (Campina Grande Federal University, Brazil), one of the developers of the CIEP (*Almeida, 2015a*).

Ranks were compiled for each of the following indices: CIEP, EPI, ESSI, LPI-NE, aENV, pENV, EFpc and EFtot, EVI, EWI. Countries were only included if there was a value for each index, resulting in a total of 114 countries. Original rankings were normalized using the following formula:

$$z_i = \frac{(x_i - \min(x))}{(\max(x) - \min(x))} * 100$$

where $z_i$ is the $i$th normalised value in the set, $x_i$ is the $i$th value in the dataset, $\min(x)$ is the minimum value in the dataset, and $\max(x)$ is the maximum value in the dataset.

Stevens et al. (2023), *PeerJ*, DOI 10.7717/peerj.16325

**Table 3** Summary of four multidimensional indices, including conceptual framework, weighting and aggregation choices, strengths and weaknesses.

| Index | Countries | Indicators | Weighting and aggregation choices | Conceptual Framework | First published in |
|---|---|---|---|---|---|
| EPI | 180 | 32 | Ex | Environmental performance –proximity to established international environmental policy goals and targets | *Esty et al. (2006)* |
| ESSI | 163 | 22 | S | Environmental state and sustainability | *Oţoiu & Grădinaru (2018)* |
| CEP | 152 | 19 | S | Assessment of environmental performance in relation to human health | *Almeida (2015a)*; *Almeida (2015b)* |
| LPI-NE | 167 | | Ex & S | Evaluation of physical environment in relation to human prosperity | *Legatum Institute (2017)* |

**Summary of indices**

**EPI**

–The EPI was preceded by the Environmental Sustainability Index (ESI) which was first published in 2000. The EPI has been published from 2006 onwards and aims to have a narrower scope than its predecessor, focusing on key environmental policy outcomes. Despite the change in name and focus, the difference between the ESI and EPI is questionable. We performed a Kendall's $\tau$ investigation of rank correlations across the different versions of the ESI and EPI and results indicate that ranks for the countries have remained similar over time (File S2).

–Issue categories are organized under three broad policy objectives (or pillars): climate change, ecosystem vitality, and environmental health.

–The environmental health objective considered a misnomer by some because it deals with human health as it is affected by the environment (*Hermele, 2009*). Together, all human health indicators receive 18% of total weight (*Wolf et al., 2022a*; *Wolf et al., 2022b*). Thus, high income countries that fare well on human health indicators, such as clean drinking water, will score more highly than low income countries.

–The EPI developing team have highlighted problems with data gaps and the need for better data collection, reporting and verification (*Wendling et al., 2018*; *Wendling, Emerson & Esty, 2020*).

–The choice of variables has been criticized because variables do not adequately capture specific dimensions of typical environmental problems in high income countries (*e.g.,* water pollution from agriculture and soil management (*Atici, 2009*, *Haberland, 2008*).

–A Joint Research Commission (JCR) analysis of the 2014 EPI found that changes in the policy objectives' weights and aggregation function led to significant variation for several countries. JCR reviews suggest that the EPI should move from expert-based weighting and aggregation methods to include multivariate analysis, uncertainty and sensitivity analysis.

**ESSI**

–Designed to build upon and guide future versions of the EPI.

–Includes human health indicators, such as life expectancy at birth, that do not reflect the aim of measuring the state of the environment.

–In contrast to the EPI, the ESSI uses factor analysis rather than expert opinion and judgement for indicator weighting.

–Country ranks generally comparable with the EPI, however, there are large differences for several countries.

**CIEP**

–Uses the Driving-Force-Pressure-State-Exposure-Effect-Action (DPSEEA) framework and aims to link health, environmental, and economic development issues.

–As the CIEP is designed to link environmental problems with human health, indicators such as access to drinking water, sanitation, child mortality, and per capita income are included.

**Table 3** (*continued*)

–Constructed using statistical techniques: data is transformed using Yeo-Johnson transformation, weighting and aggregation are performed according to CRITITC (Criteria Importance Through Intercriteria Correlation), and aggregation is linear in two steps.

–The theoretical framework and statistical methods are clear and transparent. However, the name of the index is inconsistent with the constructors' goal of linking environmental problems with human health.

–The CIEP may be useful for decision makers seeking to reduce the impacts of environmental problems on human health, however, it is less useful as an indicator of the quality of natural systems.

LPI-NE

–The LPI-NE is part of the wider Legatum Prosperity Index that uses 12 pillars of prosperity (safety and security, personal freedom, governance, social capital, investment environment, enterprise conditions, market access and infrastructure, economic quality, living conditions, health, education, and natural environment).

–Includes 6 elements: preservation efforts, oceans, forest, soil, exposure to air pollution, and emissions.

–Human health measures (*e.g.*, exposure to fine particulate matter) are included in the LPI-NE; this does not fit with the aim of evaluating aspects of the physical environment, and these indicators might be better placed under the health pillar of the wider LPI.

–Uses population-weighted scores to capture effects on individuals rather than countries. Weights are determined by two factors: (1) the relevance and significance of the indicator to prosperity (informed by academic literature and expert opinion), and (2) the statistical significance of the indicator to economic and social wellbeing of a country.

**Notes.**

Accounting based = A, Equal weighting = E, Expert opinion-based = Ex, Statistics based = S.

Abbreviations: CIEP, Composite Index of Environmental Performance; EPI, Environmental Performance Index; ESSI, Environmental State & Sustainability and Index; LPI-NE, Legatum Prosperity Index Natural Environment Pillar.

Stevens et al. (2023), *PeerJ*, DOI 10.7717/peerj.16325

Peer

**Table 4  Summary information for six environment-only indices including conceptual frameworks, weighting and aggregation choices, strengths and weaknesses.**

| Index | Countries | Indicators | Weighting and aggregation choices | Conceptual framework | First published in |
|---|---|---|---|---|---|
| EVI | 192 | 50 | E | Environmental vulnerability | *Kaly et al. (1999)* |
| EWI | 180 | 52 | E | Environmental quality –proximity to goal (low level of ecosystem stress) | *Prescott-Allen (2001)* |
| EF tot | 178 | 16 | A | Ecological deficit/ecological reserves (total) | *Rees (1992)* |
| Ef pc | | | | Ecological deficit/ecological reserves (per capital) | |
| aENV | 178 | 12 | S | Environmental impact –performance with respect to absolute (total) global impact | *Bradshaw, Giam & Sodhi (2010)* |
| pENV | | | | Environmental impact –performance with respect to total resources available per country | |

**Summary of indices**

**EVI**

–Intended to be used alongside separate economic and social vulnerability indices and designed to provide insight into the processes that can negatively influence sustainable development.

–Considers both natural and human induced changes (*e.g.*, habitat fragmentation, overfishing, earthquakes).

–Human health and socioeconomic indicators are explicitly excluded from the EVI. Using independent environmental, economic, and social vulnerability indices allows relationships between the three pillars of sustainability to be identified.

–Could be improved upon if statistical measures were used for weighting and aggregation rather than simply using unweighted averages.

**EWI**

–Measures progress towards the goal of a high level of ecosystem wellbeing and was developed to be used alongside the Human Wellbeing Index (HWI). The EWI and HWI are used to compare human conditions with the state of the environment.

–Explicitly excludes human health indicators and indicators of environmental policies and practices. This allows trade-offs between the different pillars of sustainability to be identified.

–Uses unweighted averages to build separate indices for land, water, air, species and genes, and resource use. The separate indices are then aggregated to form the EWI.

–The inclusion of statistical methods would improve the reliability of the EWI, rather than reliance on unweighted averages.

**EFtot and EFpc**

–EF accounting focusses on the supply and demand of nature. On the supply side, biological capacity is the ability of an ecosystem to produce useful biological materials and absorb $CO_2$. On the demand side, the EF measures the ecological assets a population requires to produce the resources it consumes and to absorb its waste.

–Designed to raise awareness about the impact of humans on natural systems, with the aim to guide sustainability policy.

–The Global Footprint Network have openly recognized limitations to the EF. The EF is not a full metric of environmental sustainability because some key environmental impacts are not captured. For example, the EF uses the size of fishing grounds, but does not take into account the quality of these resources (a polluted fishing ground will be less productive than a health well-managed fishing ground).

–The EF overestimates biocapacity because it does not distinguish between sustainable and unsustainable land use.

–The EF has been criticized because of its fixed weighting scheme where some categories receive identical weights. When identical weights are assigned the potential interlinkages of the various components are ignored. Ideally, multivariate analysis should be used to explore the nature of datasets and assess the contribution of the different variables.

**Table 4** (*continued*)

–Despite its limitations, footprint accounting can be a powerful tool to visualize human exploitation of resources in a single dimension (the land and water area required to support human use). For example, one international study (covering Australia, Canada, United States, United Kingdom, Mexico, France, Italy, Germany, Brazil, and the Netherlands) surveyed individual users on their experience and perceptions of the EF calculator. 91% of the 4,245 participants considered the EF to be a valuable tool for knowledge generation, and 78% found it useful to motivate actions (*Collins et al., 2020*). This indicates that the EF has been useful in terms of raising awareness about the impacts of humans on natural systems.

aENV and pENV

–Environmental impact variables include natural forest loss, habitat conversion, marine captures, fertilizer use, water pollution, carbon emissions, and biodiversity threat.

–The constructors acknowledge that they could not include all major indicators of environmental degradation (*e.g.*, bush meat harvest, coral reef habitat quality, freshwater degradation) because data was not available at the global scale.

–The conceptual framework clearly states what the index is measuring (environmental impact) and explicitly excludes confounding human health and socioeconomic factors to provide the most accurate assessment of impact for natural environments.

–The ENV uses multivariate statistical analysis to account for interlinkages of the various components and to determine component weights.

–Of the two versions, the aENV is considered a better reflection of a country's contribution to the global environmental state.

**Notes.**

Accounting based = A, Equal weighting = E, Expert opinion-based = Ex, Statistics based = S.

Abbreviations: EWI, Ecosystem Wellbeing Index; EF tot, Ecological Footprint total; EF pc, Ecological Footprint per capita; EVI, Environmental Vulnerability Index.

**Table 5** A. Degree of Correlation. B. Correlation Matrix for EPI (2020), ESSI (2018), CIEP (2015), LPI-NE (2021), EVI (1999), EWI (2001), EF tot (2018), EF pp (2018), aENV (2010), pENV (2010).

**(a) Degree of correlation.**

| moderate negative | weak negative | very weak negative | very weak positive | weak positive | moderate positive | strong positive |
|---|---|---|---|---|---|---|
| -0.59 to -0.40 | -0.39 to -0.20 | -0.19 to -0.01 | 0.00 to 0.19 | 0.20 to 0.39 | 0.40 to 0.59 | 0.60 to 0.79 |

**(b) Correlation Matrix for EPI (2020), ESSI (2018), CIEP (2015), LPI-NE (2021), EVI (1999), EWI (2001), EF tot (2018), EF pp (2018), aENV (2010), pENV (2010).**

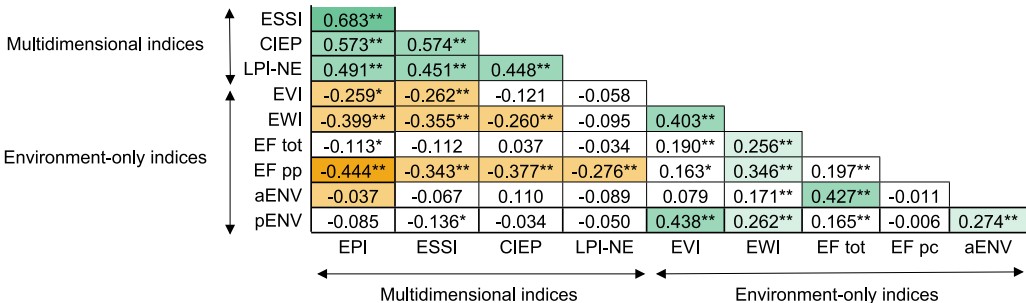

**Notes.**

*Correlation is significant at the 0.05 level (2-tailed). **Correlation is significant at the 0.01 level (2-tailed).

Abbreviations: CIEP, Composite Index of Environmental Performance; EWI, Ecosystem Wellbeing Index; EF tot, Ecological Footprint total; EF pc, Ecological Footprint per capita; EPI, Environmental Performance Index; ESSI, Environmental State & Sustainability and Index; EVI, Environmental Vulnerability Index; LPI-NE, Legatum Prosperity Index - Natural Environment Pillar.

To determine the degree of agreement between ranks we used Kendall's $\tau$ coefficient, which is a commonly used measure of rank correlation (*Field, 2018*). Because several ranks were tied, we chose Kendall's $\tau$ coefficient over Spearman's rank correlation because it is considered more appropriate in the case of tied ranks in the data set (*Field, 2018*). Kendall's $\tau$ was calculated using IBM SPSS Statistics (Statistical Package for Social Sciences) for Windows, Version 28. If these distinct environmental indices assess similar measurable aspects of environmental features of countries, moderate-to-high positive correlations are to be expected.

Second, we investigated rank correlations for indices that were published in the same year. Original ranks were normalized using the method outlined previously and Kendall's $\tau$ coefficient was used to determine the degree of agreement between ranks for: the pENV and EPI (2010), the aENV and EPI (2010), the EFpc and EPI (2018), EFtot and EPI (2018).

Third, we examined rank correlations for pillars within the EPI. For the 2010 version of the EPI, the correlation between the environmental health and ecosystem vitality pillars could be tested because the environmental health pillar consists exclusively of human health and welfare indicators, while the ecosystem vitality pillar consists exclusively of indicators of ecosystem pressures, states, and impacts. For the most recent versions of the EPI, human health and welfare indicators are no longer the sole components of the environmental health pillar. Thus, a simple correlation analysis for the pillars is not possible.

Finally, we compiled rank summaries for four countries to investigate rank patterns for low and high income countries. Germany and New Zealand represent high income countries in the Europe and Pacific regions, respectively. The countries of Mongolia and Niger represent low income countries in Asia and Africa, respectively. The four countries face a range of different environmental issues. Germany has a high emissions profile and faces issues with air and water pollution as well as biodiversity loss (*European Environment Agency, 2023*; *OECD, 2023*). Environmental issues in New Zealand include water pollution, soil degradation, biodiversity loss, and a high emissions profile (*OECD, 2017*; *for Environment, 2022*). In Niger, environmental issues include land degradation and desertification, pressure on natural resources from population growth, and pollution from mining (*Larsen & Mamosso, 2013*; *The World Bank, 2021*). Environmental issues in Mongolia include habitat and biodiversity loss, and pollution from mining (*Rossabi, 2021*).

## Results

If these distinct environmental indices assess similar measurable aspects of environmental features of countries, moderate-to-high positive correlations are to be expected. However, we found positive correlations between multidimensional indices, positive or no correlations between environment-only indices, and negative or no correlations between multidimensional and environment-only indices.

### Rank correlations for ten national level indices

Correlations ranged from moderate negative to strong positive (Table 5A and 5B). The highest positive correlation was between the EPI and ESSI, which indicates that the increase in a country's EPI rank was associated with an increase in ESSI rank. There were moderate positive correlations between the EPI and LPI-NE, EPI and CIEP, CIEP and ESSI, CIEP and LPI-NE, ESSI and LPI-NE, aENV and EFtot, pENV and EVI, EVI and EWI. There were also weak positive correlations between the EFpc and EWI, EFtot and EWI, aENV and pENV, EWI and pENV.

There was a moderate negative correlation between the EPI and EFpc. This negative correlation indicates there was an inverse relationship between the two indices, *i.e.,* the increase in a country's rank in the EFpc was associated with a decrease in rank in the EPI (or *vice versa*). There were also weak negative correlations between the EFpc and ESSI, Efpc and CIEP, aENV and EPI, EWI and EPI, EFpc and LPI-NE, EWI and ESSI, EVI and ESSI, EWI and CIEP, and EVI and EPI.

### Rank correlations for indices published in the same year

There was a moderate negative correlation between the 2018 versions of the EFpc and the EPI (Kendall $\tau = -0.483$, $P < 0.01$). Ranks varied up to 92 places (normalized 0–100). There was a very weak negative correlation between country ranks for the 2018 versions of the EFtot and the EPI (Kendall $\tau = -0.109$, $P < 0.05$) and country ranks varied by up to 93 places (normalized 0–100) (File S1).

There was a very weak negative correlation between country ranks for the 2010 versions of the aENV and EPI (Kendall $\tau = -0.072$, $P < 0.05$) and country ranks varied up to 86 places (normalized 0–100). There was also a very weak negative correlation between ranks

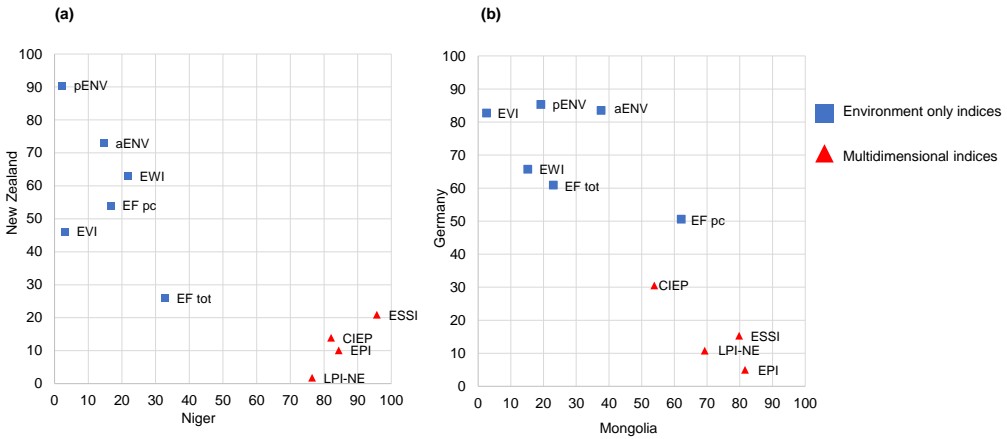

**Figure 2** **Relation between: (A) ranks for New Zealand and Niger, (B) ranks for Germany and Mongolia (relationship break down by category of environmental index).** Note: Abbreviations: Composite Index of Environmental Performance (CIEP); Ecosystem Wellbeing Index (EWI); Ecological Footprint tot al (EF tot); Ecological Footprint per capita (EF pc); Environmental Performance Index (EPI); Environmental State & Sustainability and Index (ESSI); Environmental Vulnerability Index (EVI); Legatum Prosperity Index Natural Environment Pillar (LPI NE).

for the 2010 versions of the pENV and EPI (Kendall $\tau = -0.156$, $P < 0.01$) and country ranks varied up to 99 places (normalized 0–100) (File S1).

### Rank correlations for pillars within EPI (2010)

There was a weak negative correlation between the Ecosystem Vitality and Environmental Health pillars of the 2010 version of the EPI (Kendall $\tau = -0.219$, $P < 0.01$). This indicates that an increase in a country's environmental health ranking was associated with a decrease in its ecosystem vitality ranking (or *vice versa*) (File S1).

### Summary of rankings for Germany, New Zealand, Mongolia and Niger

When normalized 0–100, New Zealand's rank varied by up to 88.6 places (Fig. 2A) and Germany's rank varied by 79 places (Fig. 2B). Germany and New Zealand achieved the highest scores in multidimensional indices, and the lowest scores in environment-only indices. Niger's rank varied by up to 94 places (Fig. 2A), while Mongolia's rank varied up to 80.3 places (Fig. 2B). Mongolia and Niger achieved the highest scores in environment-only indices, and the lowest scores in multidimensional indices.

Of note, when considering weighting methods there was no clear pattern across the different indices. For example, rankings for New Zealand and Niger were quite disparate in the ESSI and pENV (20.9 and 90.4; 95.7 and 2.3 respectively), even though both indices use statistics-based weighting procedures. However, when considering Germany, New Zealand, Mongolia, and Niger's rankings across multidimensional and environment-only indices clear patterns emerge. Mongolia and Niger scored well in environment-only indices, while New Zealand and Germany scored well in multidimensional indices.

## DISCUSSION

Composite indices have been widely used to rank the environmental performance of nations. Despite their benefits, environmental indices can have numerous drawbacks; poorly constructed or communicated indices can undermine efforts to identify and address environmental failings and misguide policy messages and decisions (*Böhringer & Jochem, 2007*; *Bradshaw, Giam & Sodhi, 2010*; *Haberland, 2008*; *Morse & Fraser, 2005*). This review sought to answer three questions. The first question asked: which conceptual framework(s) or indicators inform the development of country rankings in environmental indices? We split environmental indices into two broad categories according to their conceptual frameworks and the indicators included. Multidimensional indices measure environmental performance in relation to human health and welfare functions, and typically incorporate indicators such as human access to safe sanitation, exposure to pollutants, and per capita income. Environment-only indices, on the other hand, focus exclusively on functions of natural capital that represent pressures, states, or impacts and explicitly exclude indicators related to human health and welfare. Both multidimensional and environment-only indices use a range of weighting and aggregation methods.

Second, we sought to understand the impact of different conceptual frameworks (and indicators) on global rankings. We examined rank correlation for ten national level environmental indices, and where possible, we investigated rank correlations for indices published in the same year. While previous research has demonstrated the sensitivity of rankings to different weighting methods (*Halkos & Zisiadou, 2018*; *Pinar, 2022*), in our analysis we did not observe any clear patterns across the indices. Our results indicate the following patterns: (1) multidimensional indices (CIEP, EPI, ESSI, LPI-NE) were positively correlated with one another, (2) environment-only indices (EFpc, EFtot, aENV, pENV, EVI, EWI) were positively correlated with one another, or there is no correlation, and (3) multidimensional indices were either negatively correlated with environment-only indices, or there is no correlation. Researchers have used a range of weighting methods. The multidimensional indices included in this study showed much stronger positive correlations with each other, than do environment-only indices. This may be due to similarities in the indicators and datasets used. The ESSI, for example, was developed to guide revisions of the EPI and uses similar indicators but with a statistics-based weighting system to account for relationships between them. However, the general pattern of correlations observed for multidimensional and environment-only indices indicate that conceptual framework and indicator choice may influence a country's rank among different indices.

As a complement to our analysis of rank variation for existing national-level indices, we examined rank correlations within the EPI's pillars. The ecosystem vitality pillar includes indicators that fall under the environmental pillar of sustainable development, whereas the environmental health pillar includes indicators that fall under the social and economic pillars of sustainability. The negative correlation between the environmental health and ecosystem vitality pillars of the 2010 version indicates that an increase in a country's environmental health rank is associated with a decrease in the ecosystem vitality ranking. Our findings support previous research (*Bradshaw, Giam & Sodhi, 2010*) and
demonstrates that environmental sustainability can be distinguished from social and economic sustainability. This highlights how the inclusion of human health and welfare indicators may influence a country's overall rank with the potential to misrepresent the environmental sustainability of different countries. The environmental health pillar of most recent version of the EPI includes indicators relating to waste management as well as human health and welfare indicators. Thus, it is not straightforward to compare ranks for the environmental health and ecosystem vitality pillars. While it is beyond the scope of this article to disaggregate and re-weight the indicators and issue categories of the most recent version of the EPI, this could be explored in future research.

One limitation in our correlation study is that the release dates for the indices range over several years. Unfortunately, some indices only have one iteration, and it was therefore not possible to conduct a year-on-year comparison for every index. While this is an important limitation, analyses of indices such as the EPI have confirmed strong positive correlations between ranks over multiple iterations (File S2). This indicates that country ranks within the same index do not vary markedly over time and provides some assurance in the results reported here. In particular, the EPI (and other indices such as the LPI-NE) could be constructed without indicators related to human health and welfare, and have ranks compared to those in existing environment-only indices, such as the Ecological Footprint. As the EPI uses expert opinion-based weighting, different weighting procedures (*e.g.*, multivariate analysis) could also be considered along with sensitivity analysis. Additionally, subsets of data for the same years could be analysed for a more direct comparison.

Thirdly, we sought to understand the extent to which the different environmental indices capture environmental problems in low and high income countries. We have used the examples of Germany, New Zealand, Mongolia, and Niger to illustrate the complexities involved when interpreting ranks among different indices. Low income countries, such as Mongolia and Niger, often score poorly in multidimensional indices and well in environment-only indices. In Niger 42.9% of the population live in extreme poverty (*The World Bank, 2021*) and only 13% of the population has access to basic sanitation services (*UNICEF, 2020*). In Mongolia, 28% of the population lives in poverty (*The World Bank, 2021*) and only 27% of the population has access to sanitation (*UNICEF, 2017*). Thus, the inclusion of indicators such as access to sanitation and GDP in multidimensional indices complicates interpretation, and ultimately leads to a less optimistic view of Mongolia and Niger's environments when compared to high income nations. By contrast Germany and New Zealand both score well in multidimensional indices. Germany and New Zealand are classified as a high income nations and this is reflected in indicators such as access to clean drinking water and sanitation (*UNDESA, 2021*; *World Bank, 2023a*; *World Bank, 2023b*; *World Bank, 2023c*). The inclusion of such indicators provides a more optimistic view of German and New Zealand's environments when compared to lower income nations. Because interactions between economic growth, improvements in human health and welfare, and environmental quality are complex, it is important to treat multidimensional indices with caution—multidimensional indices require more layers of interpretation than environment-only indices.

Research looking at rank differences between environmental indices and socio-economic has largely focussed on relationships between the EPI and GDP per capita, human development (*Lai & Chen, 2020*; *Tektüfekçi & Kutay, 2016*; *Wolf et al., 2022a*; *Wolf et al., 2022b*; *Wurie & Pillai, 2014*). While the scope of this article was concentrated on examining correlations in environmental indices, the dataset collated for the present research could be utilised to extend research into correlations between multidimensional and environment-only indices and other composite indices or indicators of human health and welfare such as GDP. These may include sustainability and climate indices such as the HDI, Global Sustainability Index (GSI), Sustainable Development Index (SDI), and Global Social Mobility Index.

Our article provides insight into the benefits and challenges of constructing environmental indices, which can help inform the future design of (improved) indices. It is important to assess whether the conceptual framework and indicators included are appropriate. If an index is intended to measure the qualities or functions of natural capital linked to environmental as well as human health and welfare or policy processes, then it may be appropriate to include indicators such as access to sanitation, indoor air pollution, and per capita income. If the index is intended to measure the environment-only, then functions linked to human health and welfare, or policy should be excluded. If such indicators are included, the actual state of natural environments may not be accurately represented.

In this article we have: (1) examined existing environmental indices and quantified rank variations, (2) identified patterns across multidimensional and environment-only indices, and (3) examined how conceptual frameworks and methodological choices influence global rankings. Our findings highlight wide variations in country ranks among existing environmental indices. Specifically, our results suggest that conceptual framework and indicator choice can affect ranking and produce a more, or less, optimistic view of a country's environment. While weighting procedures and sensitivity analysis may have some impact on overall ranking, it is likely that conceptual framework and indicator choice have a larger effect.

## ACKNOWLEDGEMENTS

A significant contribution to this article was made by Dr Lynda Petherick, who sadly passed away during the preparation of the manuscript. We dedicate this article in her memory. We would like to thank the reviewers for the valuable feedback provided, which helped us to improve our manuscript.

### Funding

This research was supported by a Victoria University of Wellington Doctoral Scholarship (Shelley M. Stevens). The funders had no role in study design, data collection and analysis, decision to publish, or preparation of the manuscript.

## Grant Disclosures

The following grant information was disclosed by the authors:
Te Herenga Waka Victoria University of Wellington Doctoral Scholarship.

## Competing Interests

The authors declare there are no competing interests.

## Author Contributions

- Shelley M. Stevens conceived and designed the experiments, performed the experiments, analyzed the data, prepared figures and/or tables, authored or reviewed drafts of the article, and approved the final draft.
- Michael K. Joy conceived and designed the experiments, performed the experiments, analyzed the data, authored or reviewed drafts of the article, and approved the final draft.
- Wokje Abrahamse conceived and designed the experiments, performed the experiments, analyzed the data, authored or reviewed drafts of the article, and approved the final draft.
- Taciano L. Milfont conceived and designed the experiments, performed the experiments, analyzed the data, authored or reviewed drafts of the article, and approved the final draft.
- Lynda M. Petherick conceived and designed the experiments authored and reviewed drafts of the article.

## Data Availability

The raw data for our correlation analysis and the summary results of the rank correlation we performed for 10 iterations of the ESI/EPI are available in the Supplementary Files.

## Supplemental Information

Supplemental information for this article can be found online at http://dx.doi.org/10.7717/peerj.16325#supplemental-information.

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
