# Peer review of "Composite environmental indices—a case of rickety rankings"

_PeerJ, doi:10.7717/peerj.16325_

## Round 0.1 · original submission · Major Revisions

The reviewers had good things to say about your submission but thought that it could be made better. Please consider their comments in revising the manuscript.

Reviewer 1 ·

Basic reporting

The paper seeks to contribute to a more rigorous use of environmental indices, a topic that fits within the scope of the journal. The topic addressed is very important in a setting in which the number of environmental indices available is increasing considerably. Having said this, I have mixed feelings about the paper. I think the motivation is sound and that it can be a relevant paper in the field, but it does not yet deliver on its promises. For instance, the design and content of the review needs to be improved, the methodology needs to be tweaked so that the key research questions can be answered more adequately, some of the assertions made need to be rethought, etc. Thus, I feel the paper needs to be revised substantially before it can be considered for publication. As a result, my recommendation is major revisions.

In the following lines, I make several suggestions to improve the paper (along with other comments) hoping that they will help the authors improve their work. Of course, it is up to them what to take on board. I also include several references. Some of them refer to papers that you might have overlooked, but most of them are used to illustrate some of my points. The latter should not be included in your text unless they are relevant.

Specific comments

Title
At this point, I feel that the title does not reflect the content of the paper. For that to happen, I think the authors should put more emphasis on how environmental indices have been used so far, and on how the have been misused. For instance, there is only one example (Rowarth) for the latter case.

Introduction
The introduction is too long and through many of the examples provided assumes that the reader is familiar with various indices that are only properly introduced in the results section. I would welcome a more targeted introduction that clearly sets the context, the research questions and the novelty of the paper. Thus, I would recommend you to rethink what you want to include and rewrite it accordingly.

L52-54: This is at the core of the problem the paper tries to address, but it remains very superficial and abstract in the introduction. You have used the example of New Zealand in the discussion, which I would summarise here without being as specific as you have been in that section. Having additional examples would strengthen the problem you try to address. I would argue that these do not necessarily have to be related to the environment. For instance, I have seen a few papers addressing this topic in relation to university rankings (e.g. Saisana et al. 2011), but there are probably other fields as well.
Saisana, M., d’Hombres, B., & Saltelli, A. (2011). Rickety numbers: Volatility of university rankings and policy implications. Research policy, 40(1), 165-177.

L55-56: I think this sentence is a bit problematic and partly reflects some of the problems I see with the paper. You refer to inconsistencies between indicators, which I feel it implies that environmental indices should strive to describe a very particular system (from the text I think it is the state of the environment). Nonetheless, environmental indices do not intend to describe a single and absolute system. By definition, there is nothing wrong or inconsistent with including indicators of environmental policies, environmental pressures or impacts in an environmental index. In other ways, the environmental dimension of sustainable development can cover many different subsystems, one of which is the state of the environment. The key bit in this case is to choose indicators that are consistent with the definition of the system that is to be described. I am not saying that you have the position I have described first, but it comes across that way in parts of the text. Here I would ask you to be a bit careful with the terminology you use (an issue you include in Table 1) when referring to environmental systems, environmental sustainability, the environmental dimension of sustainable development and similar concepts.

L64-67: While there are some steps that are common in all indices (normalisation, weighting and aggregation), others are not. For instance, not all the indices have or document the conceptual framework (Kwatra et al. 2020).
Kwatra, S., Kumar, A., & Sharma, P. (2020). A critical review of studies related to construction and computation of Sustainable Development Indices. Ecological Indicators, 112, 106061.

L70: The term ‘ranking goals’ sounds odd. I would encourage you to find a different term. Indices do not necessarily intend to create rankings.

Table 1: Is the content of the table supposed to be comprehensive? You use this table to guide part of your argumentation later on, but it feels incomplete. For instance, for the theoretical framework you argue that indices can be split between those that focus exclusively on the environmental pillar of sustainability and those that go beyond it. I have issues with this statement (in line with a previous point I have made), but I will first argue that there are other ways to split indices. A very common one refers to the weak and strong sustainability perspectives (see Usubiaga-Liaño and Ekins 2021 for a brief review based on that criteron). Of course, there might be others.
Usubiaga-Liaño, A., & Ekins, P. (2021). Time for Science-Based National Targets for Environmental Sustainability: An Assessment of Existing Metrics and the ESGAP Framework. Frontiers in Environmental Science, 524.

Another way of sorting indices could be if they provide information for a single year or if they show trends. I have noticed that the latter is increasingly being part of SDG assessments (e.g. Eurostat 2020).
Eurostat. 2020. Sustainable development in the European Union. Monitoring report on progress towards the SDGs in an EU context. Luxembourg: Eurostat.

Table 1: A second point on this table is the terminology. As I have argued before, the environmental dimension of sustainable development is quite broad and can encompass many different subdimensions. Here you refer to the environmental pillar of sustainability, which I assume you use to mean something narrower. If that is the case, please define it in the introduction so that the terminology is clearer throughout the text (you have some definitions of environmental sustainability in the previous reference).

Table 1: I agree with what it seems to be your initial hypothesis: that the inclusion of human health and policy indicators affects index scores and rankings. The problem I have is that the hypothesis does not seem to be the result of a logical conclusion. It comes out of the blue in this Table and in the text.
After elaborating on the potential misuse of (environmental indices), you could elaborate on the gap between policy adoption and implementation, on why you could expect different country performances in health and environmental indicators, which would set the ground for your hypothesis.

Table 1: The issue of indicator choice needs to be elaborated more. Ideally, developers should use specific criteria to select indicators. Only by having clear criteria you can unsuitable indicators are chosen and key indicators excluded (as it is stated in Table 1). In this case, the ‘relevance’ criterion is of major importance (e.g. Hak et al. 2016 in the context of the SDGs). As an example, Usubiaga-Liaño and Ekins 2021 provides specific criteria that indicators need to meet to be relevant in the context of the framework they use.
Hák, T., Janoušková, S., & Moldan, B. (2016). Sustainable Development Goals: A need for relevant indicators. Ecological indicators, 60, 565-573.
Usubiaga-Liano, A., & Ekins, P. (2021). Monitoring the environmental sustainability of countries through the strong environmental sustainability index. Ecological Indicators, 132, 108281.

Table 1: In the last row, I would also refer to the normalisation process.

L70-onwards: The text assumes that the reader is familiar with many of the indices used to give examples. Keep in mind that the examples are very specific and that the main information about the indices is given in the results section (Table 2, I believe). Please rethink how you want to present the information. My feeling is that the paper does not provide a very thorough review of the indices (instead I would call it overview), which makes me wonder to which extent this is needed in light of the various reviews that already exist (e.g. Kwatra et al. 2020, Usubiaga-Liaño and Ekins 2021).
Kwatra, S., Kumar, A., & Sharma, P. (2020). A critical review of studies related to construction and computation of Sustainable Development Indices. Ecological Indicators, 112, 106061.
Usubiaga-Liaño, A., & Ekins, P. (2021). Time for Science-Based National Targets for Environmental Sustainability: An Assessment of Existing Metrics and the ESGAP Framework. Frontiers in Environmental Science, 524.

If you think the review is important, I would recommend you to further develop the criteria used and to highlight more clearly the key insights you get from it. Given that the main point of your paper is based on the correlation analysis, an overview such as the one you given in Table 2 is necessary, but I would probably present it as earlier in the document (not in the introduction though); perhaps before the methods. In this line, I agree that you should use indices with data for many countries in the correlation analysis, but I do not see a reason to exclude them from the review (again, if you decide to do one).

L74-75: This sentence is repeated (see L70-71).

L84-86: This needs a reference.

L89: Is “correlate” the right term?

L91-94: “unsuitable” is a very strong term. The choice of indictors needs to balance aspects related to relevance, methodological soundness, data quality, easiness to communicate and many more. There might be good reasons for CIEP and EPI to choose an access indicator instead of a quality indicator. While the latter seems more relevant, there might be issues related to data availability, comparability or quality that made them choose the first one. Keep in mind that those indices are constructed for many countries. Of course, if you were to compute that for a single country or region (as in the case of the IHOBE reference you use), there might be other indicators that are more adequate, but from a global perspective, there might not be. Thus, I would encourage you to be more cautious with some of the assertions you have in the text. In case it is of any help, I will highlight other assertions that I think could benefit from a more careful wording.

L99: “questionable” - strong assertion that I think it is not justified. There might be better data, but “questionable” is a strong term.

L101.102: In my experience, one you get into the metadata of indicators, caution applies to more indicators than what it is usually expected.

L120: The issue of terminology comes again. Please check the first EPI report in which they outline the main differences between ESI and EPI. You will see that EPI is not intended to reflect sustainability, but rather environmental outcomes linked to policy goals.

L140-141: I am not familiar with ESSI, but I imagine that, beyond a different weighting system, it will have different indicators compared to EPI. Because of this, you should have better examples to make the point about how weights influence rankings. Any uncertainty analysis of an environmental index should suffice.

L142: “variable” or “indicator” choice?

L152-onwards: The results of Bradshaw et al. are reported in too much detail.

L177-178: Simply expanding the work of Bradshaw does not seem to be novel enough to me. You can add novelty by elaborating on the misuse of indices and by testing your initial hypothesis more thoroughly.

Experimental design

I have already commented on the weaknesses I see in the review, so I will not do it again here. Instead, I will make you points about the correlation analysis.

First, you could also use the Spearmann rank correlation analysis, which is very intuitive and easy to implement. Second, I am not sure if the way you test your hypothesis is the best one. While you test the correlation between indices that include (or do not include) policy and human health aspects, I wonder if it would make sense to complement this analysis by testing the correlation between a subindex calculated with environmental state indicators, and a subindex calculated with policy and human health indicators. For instance, calculating an EPI subindex of human health vs an EPI subindex of environmental state. Whether this makes sense will depend on how many indicators there are in each subindex and in other factors, but it feels like a good idea to me.

L242-243: These two indices have different years: 2017 and 2018. The Ecological Footprint has data for 2018 as well.

Validity of the findings

Results
I will not say much about this section, as I expect it to change based on my previous suggestions.

L275: I think this should be Table 3.

L310: While I think it is a good idea to illustrate some of your points using high-income and low-income countries, I do not think this item warrants to be termed a case study, since it only covers a single paragraph. In any case, this might just be a minor wording issue.

Discussion
This section makes important points, but will need to be revised based on previous comments.

L329: I do not think you are quantifying ‘inconsistencies’. So far, I do not have the impression that you are dealing with inconsistencies, but with indices that try to measure different things or that they simply use different indicators.

L332: As I mentioned before, I do not think many of the indices are trying to solely describe the state of the environment, so I am not sure to which extent they provide a ‘misleading view’.

L346-349: This reinforces the need to have proper definitions somewhere in the text. While I agree that many environmental indices do not represent well the state of the environment, they might not have the intention to do so, or they might want to do more than that.

Conclusions
L428-430: I do not think you can state this. Environmental indices should not solely focus on the environmental state. As I argued before, there is much more they might want to do.

Additional comments

Other
The references appear twice in the text.
What are the figures for? I have not seen any references to them in the text.

Reviewer 2 ·

Basic reporting

Writing:
Overall, clear and unambiguous English was used throughout the manuscript. To provide a more succinct and impactful review, the authors could shorten much of the writing throughout the manuscript. The idea about the discrepancy between the two broad types of environmental indices (those that are multidimensional because they include human health, socioeconomic, and policy indicators, and those that are more strictly environmental indicators) were repeated too often throughout the manuscript. The introduction of 9 paragraphs (as best as I could count, given the awkward formatting) would be better off reduced to about 5-6 paragraphs by being more selective on which details to include. Alternatively, because it is a review, the introduction could remain long, and include subsections to more effectively convey the background information to readers. Similarly, the Discussion and Conclusions could be combined and shortened. The manuscript, in its current state, is relatively straightforward in its analysis and findings, and thus does not need a conclusion section in addition to the discussion.



Background:
Literature references and generally sufficient field background/context were provided.



Article structure:
The article was structured in a traditional format of Abstract, Introduction, Materials & Methods, Results, Discussion, Conclusions, and References. Raw data was shared. However, there was minor mis-numbering in references to Tables, and no Figures were referenced in the manuscript, even though the authors submitted three figures. The first supplemental file was also never referenced in the manuscript. The Results section could be rewritten in a way that provides more meaning to the study results (e.g., strongest correlations were…) rather than lists of correlations grouped into three paragraphs. The information on correlations and p-values are already provided in Table 3. The text should provide more information and interpretation of these results.

Other minor edits could help readers more quickly understand and remember the information in the manuscript. For example, all indices could have acronyms, rather than some spelled out throughout the manuscript, especially when all of them have acronyms in the tables and figures. Please stay consistent. Also, the authors could use a much briefer naming for the two main types of indices. I suggest something along the lines of “multidimensional” and “environment only”. Once defined in the introduction and/or methods, the authors would not have to repeat “human health, socioeconomic, and policy indicators”.



Broad and cross-disciplinary interest:
The review is broad, has cross-disciplinary interest, and is within the scope of the journal because it is a review of environmental indices, including human health, socioeconomic, and policy indicators for countries around the world.

Another review paper in this field was published relatively recently, but does not provide the same study goals and analysis, and is not as accessible to broad/diverse audiences as the current manuscript.

El Gibari et al. 2018. Building composite indicators using multicriteria methods: a review. J Bus Econ. https://doi.org/10.1007/s11573-018-0902-z



Study subject and audience:
The introduction could be better refined to make the goals/objectives of the review clearer. The authors state in the last paragraph of their introduction, the aim of the paper, which is worded in a way that reads as an unbiased, search for a particular result. Instead, I suggest the authors rephrase some of their sentences to quality their review as a search for patterns, if they exist, rather than aiming to find a pattern already in mind and a demonstration. Furthermore, as I was reading the manuscript, I think there are two objectives at hand: 1) comparison of environmental indices that are multidimensional vs. environment only, and 2) comparison between averaging methods (based as quantitative/quasi-quantitative, expert opinion, accounting, equally weighted, unweighted).

The audience and application of the findings are quite clear in the introduction, but could be more succinctly worded for greater impact.

Experimental design

Journal scope:
The article content is within the Aims and Scope of the journal.



Rigorous investigation:
The investigation was conducted thoroughly in some aspect, but was lacking in some other aspects. A thorough search for environmental indices was conducted, and the straightforwardness of analysis helped draw out patterns with some confidence in the findings.

I suggest revising Table 3 so that the “Degree of correlation” table is numbered 3a, and the correlation matrix is numbered Table 3b. Categories named something along the lines of “Multidimensional indices” and “Environment only indices: spanning underneath and to the side of the relevant indices would help readers quickly interpret Table 3b. The positive correlations between multidimensional indices and between environment only indices could readily be interpreted. Likewise, a quick connection between the pattern of negative correlations with the multidimensional indices vs environment only indices could be understood by readers without needing to remember what each acronym represents.

Furthermore, in Table 3, the categories for “very weak negative” and “very weak positive” should not be colored blue or red, but a neutral color like white. A correlation of 0 or close to 0, is essentially zero correlation, and trends or patterns should not be read into these findings at all. This coloring should be edited in both Tables 3a and 3b. This would help provide a more rigorous interpretation of results in the correlations between the multidimensional environmental indices (CIEP, EPI, ESSI, LNE) and the more strictly environmental indices (aENV, pENV, EFpp, EFtot, EVI). I do not think the conclusions of the review would change, but findings in this table would provide a more reliable interpretation.

The indices in Table 4 could be reordered to match the same ordering as Table 3. This would also help readers quickly see patterns that are grouped together. The low/high rankings would thus be more grouped together. The rows (or some cells) could also be color-coded to reflect the grouping of whether or not human health, socioeconomic, and policy indicators were included. A different set of colors other than blue and red would help differentiate the colors associated meaning used in Table 3. For the weighting column, I suggest using the acronyms S, Ex, A, E, U, and defining these in the caption (S=statistics-based, Ex=expert opinion, etc.).

For greater depth in the possible complexity in the multidimensional indices, I wonder about how human health, social/economic classes, and policy affect the environment over decades. Interaction effects could become more evident over time, particularly with use/depletion of natural resources and climate change impacts on people and the environment. Are there some examples of countries with poor human health, social/economic classes, and policies negatively impact the environment with compounding effects over time? While, some other countries with good human health, social/economic classes, and policies are better setup to maintain and restore environmental conditions over time? I think that some readers, including myself, would wonder about the validity of multidimensional indicators on the environment, particularly when there could be interaction effects over decades. These multidimensional indicators simply need one more layer of interpretation. This would help add to the story of “multidimensional indices” are not the same as “environment only” indices.

I suggest some minor edits in Table 2:
Define the JRC acronym (Joint Research Commission) when first encountered, and then use the acronym. Either define the weighting and aggregation choices in the table or don’t. If not, then non need for “(1:1:1…). If defined, more details needed for most of them. The number for “Indicators included” for “Legatum Natural Environment” is missing. Merged the cells with the same weighting and aggregation choices rather than using the “ symbol.

Figures 2 and 3: Please make sure these are references in the text. These figures should also be bar plots rather than linear. Plotting only the 10 largest rank differences is making a point rather than showing results in an unbiased manner. I suggest significant revisions here, or removal of these figures.



Method description:
The methods were described in a manner that could be replicated for the most part. The authors could be more detailed in how they normalized the original ranks (line 240).



Source citations:
Sources are adequately cited.



Organization of paragraphs/subsections:
The review was organized for the most part, but minimization of ideas repeated throughout the manuscript would provide more room for additional ideas and a deeper dive into the meaning of the complexity and possible interactive effects underlying the multidimensional indices.

Validity of the findings

Impact and novelty:
Although the findings are relatively straightforward and obvious, they can be quite impactful. Better organization and phrasing of the content could help layer the meaning of multidimensional indices more effectively.



Conclusions:
The conclusions are stated in multiple, but could be done more succinctly in one paragraph (i.e., the last paragraph of the discussion). The original research question(s) should be revisited, as I had suggested as a 2-part objective in the introduction; thus, the conclusions should reflect these.



Argument:
There is somewhat of a well-developed and supported argument that meets the goals set out in the introduction, but could be more refined. I found that there were a lot of details in the introduction that raised some secondary questions. I think these were phrased as part of the review, but could be better linked to the methods and results. Similarly, the discussion dove into some details that I found could be more strongly tied to the methods and results. The discussion could revisit the issues brought up in the introduction pertaining to Table 1, and in context of their methods and results.

At other times, the obvious seem to be stated. For example:

“If the goal of ranking is to account for multiple dimensions of sustainability, then the inclusion of human health, socioeconomic, and policy indicators (either directly or as a calculation component) may be appropriate. However, if the goal of ranking is to account only for the environmental component of sustainability, then human health, socioeconomic, and policy indicators should not be included, as they have the potential to confound or dilute the environmental component of sustainability.” (lines 342-348)

“To avoid confounding results, environmental indices should focus solely on the environmental domain and explicitly exclude human health, socioeconomic, and policy indicators” (428-430)

The last paragraph of the Discussion (lines 393-397) ties to the second objective, that I had suggested be more clearly stated in the introduction. The findings here I think could be more rigorously analyzed and interpreted.



Unresolved questions / gaps / future directions:
Many questions were listed, but should be phrased in a way that clearly states that these are for future research. I had thought that these were questions to consider based on the authors’ review and findings, and thus were overreaching. I suggest the last sentence of the conclusion be a broad statement relevant to the majority of readers, rather than a specific note about the New Zealand Quality Index that is currently under development. This last sentence would be better suited elsewhere in the manuscript.

---

## Round 0.2 · Major Revisions

After reviewing the comments from the reviewers of your revised manuscript, I believe strongly that the paper is worthy of publication; however, reviewer 2 had some good points that should be addressed. Please consider the comments from reviewer 2 (and reviewer 1 with minor comments). Once this is done, I will review the next version and make a decision.

Reviewer 1 ·

Basic reporting

The authors have made an outstanding job revising the paper, for which I congratulate them. I also welcome the detailed response to the comments I raised. The paper looks very promising now and I think it provides important novel contributions.

I still have some relatively minor comments that I think need to be addressed before publication.

L24: “The current issue in this field” sounds odd.
L45: “multiple measures into a single indicator” would read better as “multiple indicators into a single score”
L45: “feature” should be “features”
L92-95: This argument is very similar to the one in L78-82
L124: “marketed and nonmarketed” would read better as “with and without market value”
L127: “linked to” is used twice in the same sentence
L127: Should “Edkins” be “Ekins”?
L129: Please cite the source of the table in the appropriate place
L134-136: I would welcome a better definition of what multidimensional indices include. They do not only include social state indicators. For instance, the existence of a policy (a common indicator in the SDG index) is not a social state, although it does fall within the term health and other socioeconomic aspects that I think you use throughout the text.
L138: Not all environment-only indices focus on functions. Perhaps “aspects” would be a better term.
L139-140: Same as the comment in L134-136
L151-153: I assume that the statement is not true, but you do not justify why.
L163: “quality” should perhaps be “adequacy”
L176-181: Similar argument to the one in L154-160
L180-181: I think saying that there are inconsistent is better
L228-230: You seem to imply that weighting with statistical methods is always a better option, but these methods have shortcomings as well (an important one being that they cannot be easily communicated, which reduces trust in the index).
L233: “30 rank changes” should be “30 places”
L242: “this is successful” would read better as “it does so adequately”
L247: What is SMF?
Sections 2.2 and 2.3: I quite like section 2.1, and while sections 2.2 and 2.3 are very informative, I feel they cut the flow of the paper because of their length. My recommendation would be to expand Table 4 by including the main outcomes of your analysis and move it here. This would make this section much more digestible.

A minor comment on numbering. In the text, numbers from zero to ten are usually spelled out. You do this only sometimes, so I would recommend that you do it consistently.

Experimental design

L500-501: Please state which years you are using for each indicator somewhere in the text or in the supplementary material.
L517: “that represent policy objectives”. Not clear what this means.
L525-528: I am not sure you can draw relevant conclusions from two countries. See my recommendation further below.
L548: I would mention here that, generally speaking, you find positive correlatives between multidimensional indices, positive or no correlations between environment-only indices, and negative or no correlation between multidimensional and environment-only indices.
L565-577: As mentioned in a previous comment, I think this would fit better in section 2.
L578: I think a different representation of the figure would allow to include more countries meaningfully. For instance, in each tick of the x axis you could represent the scores of the ten indices for one country. Environment-only scores would use a different colour compared to multidimensional indices. This way it would still be obvious the bias introduced by health and socioeconomic indicators in developed and developing countries.

Validity of the findings

Nothing to comment in section 4.

Reviewer 2 ·

Basic reporting

The authors have done substantial revisions to the manuscript and it is much clearer than their previous version. Introduction and Review sections are better organized.

A minor comment about L65-68: “(it should be noted that rankings vary in other fields… ))” seems out of place. If this comment is making a different but related point, I think it should be its own sentence rather than a side comment. If this is meant for a broader audience, I think stating in several words how these indices differ would help readers better understand the point being made. They will likely not look up these references. This paragraph would flow better without this parenthetical comment because it seems to hinder more than help understanding.

I appreciate the three questions worded in L78-82. These questions may be worth repeating in the Discussion with brief answers to summarize.

Experimental design

One thing I first noticed was how the subsections were quite different in length. I think there can be more structure and balance in the subsections under the Review section. The review could be organized more logically into coherent paragraphs within each subsection. An introductory paragraph for section 2.2 on multidimensional indices could give an overview of the indices considered. That way, readers would know to expect a lengthy subsection for EPI, and shorter subsections for the other indices. I can see that the authors provided some background on motivation, description, weaknesses, etc. for each index. If the authors could revise each paragraph for clearer flow and consistency, it would improve the writing. Some of the sentences seem to be a string of ideas that could be better tied together in well-structured paragraphs. One of the most noticeable deviations, I think is L355-356, which is a standalone sentence.

Furthermore, I think elaboration is needed to support “First, we discuss how different understandings of sustainable development, environmental sustainability, and natural capital have informed environmental indices.” This is addressed in L114-128, but needs more. I think setting a clear foundation of concepts (development, society, economy, human health, environment, sustainable development) and their interrelatedness would help with interpretations of study findings. There is a Venn diagram and a Table of definitions of environmental sustainability, but these could be better discussed and tied to study findings.

The case study for comparison between Niger and New Zealand helps demonstrate patterns that support overall study findings. I think some justification is needed for why these two countries were selected, in addition to one being an undeveloped country and the other being developed. I think one or two more case studies for comparison would help readers better appreciate the negative correlations between multidimensional and environment-only indices, as well as the positive correlations within multidimensional indices and within environment-only indices.

At the very least, Figure 2 needs to be revised. This should not be a scatterplot because the x-axis has nominal categories with no particular increasing/ranking order. A barplot would be more suitable. (Drawing a regression line through the points would be inappropriate and meaningless.) Also, please have the order of indices consistent throughout the paper, in the review text, figure and tables.

One thing I did notice in Figure 2 was the seemingly positive correlation between multidimensional indices of Niger vs. New Zealand. As well, there is a flipped pattern in the values of the indices by country and multidimensional vs. environment-only. The authors may want to consider a scatterplot with x-axis being 0-100 for Niger and the y-axis being 0-100 for New Zealand. The points for multidimensional indices will clump in the lower right corner of the graph. The environment-only indices would clump in the top left quadrant for the most part. This could help demonstrate how indices for different countries are not equivalent because they are so far from the 1:1 diagonal line of the graph; and the clumping of points show that multidimensional indices and environmental-only indices differ from each other when comparing developing and developed countries. If creating a scatterplot, I suggest using different shaped symbols in addition to different colors so that readers can still easily distinguish points if viewing in grayscale.

Validity of the findings

I think the paper could be much more impactful if the authors could tie the results and findings with the fundamental concepts that they laid out earlier in the paper. I am still struggling with a clear understanding of how development ties to human health/welfare and the environment/ecosystem. It seems that there are two definitions/processes of development: one where development improves human health/welfare, and one that hinders the environment/ecosystem. I get a sense that environmental sustainability is different from economic sustainability and societal/human health/welfare, and that development ties to these three dimensions of sustainability (Figure 1) in different ways. But I think a clearer, deeper background in the review section and again in the discussion would help lay out a solid background and bolster validity of findings. This may include further elaboration in the comparison between Environmental Health and Ecosystem Vitality.

If correct and relevant, I suggest coloring in the portion of the Venn diagram that is only the “Environment” to represent what environment-only indices cover. Other parts of the Venn diagram can have a different color to represent multidimensional indices.

A few questions I’m still wondering about relate to: whether human-health/welfare-only indices (which the authors do not consider in this study) almost always are negatively correlated with environment-only indices; and whether climate change impacts and pollution negatively affect both the environment and human health/welfare. I see that the authors qualify the complexity of these indices in L631-633, but perhaps one or two more case studies of comparison would help delve deeper into these concepts with examples.

With 11 indices considered and criteria of at least 100 countries for further analysis, the authors have collated an incredible data set for this review. There is a lot of potential for analyses with these data.

With regards to the writing, the first paragraph of the Discussion needs editing. Instead of explicitly listing the aims of the study, I think summarizing the findings will be sufficient for readers to grasp the aims of the study. Thus, the sentences in L595-600 can be deleted. I think the sentence in L605-607 would be better placed before the sentence in L600-604. I suggest deleting the sentence “Researchers have used a range of weight methods” because it doesn’t seem necessary or add any additional meaning.

The second paragraph on limitations seems odd because it is so early in the discussion. This is usually one of the last paragraphs of a discussion. Alternatively, I suggest incorporating the idea of analyzing a subset of the data with the same years for more direct comparison in a later paragraph. For example, in addition to the suggestion of reconstructing indices without human welfare indicators (L624-628), comparable years can be part of the reanalysis.

Another point of the discussion could be on why the multidimensional indices are relatively highly correlated, while the environment-only indices are relatively less correlated.

I think overall, the authors have made really great, significant improvements to their manuscript. I think another round of major revisions would help solidify the paper.

Additional comments

Minor comments:

L83 I suggest “practices” in place of “practice”.
L124 “I think the authors mean “marketed” instead of “marked”.
L141 “Requirements” seems to be a strong word and quite absolute. Perhaps “Considerations” or another word would be better suited for this section title.
L159-160 Great sentence!
L179-181 Another great sentence!
L237-240 I think this sentence can be edited a bit to also bring in the importance of data quality (incl. time periods and data gaps) to better summarize the previous four paragraphs.
L333 The word “Effect” is missing in “Driving Force-Pressure-State-Exposure (DPSEEA)”/
L339 The incorporation of “Effect” is missing here too.
L439 I suggest removing the word “While” and just starting with “The EWI…” A new sentence can begin with “We have included…” too.
L447-449 These questions do not have to be in separate bullet points but can be in-line after the colon.
L494 Define EFpc and EFtot. I assume these mean percent and total, but am not sure.

The results section is often worded in present tense, but not consistently. I suggest the authors still use active tense, but past tense to report their findings with data from the past that may not necessarily hold indefinitely. As the authors mentioned, the results are dependent on the years examined.

Table 4 typos: “(EPI)” should be a year, I think. The names and definitions for EFtot and EFpc are reversed. The same definition is used for aENV and pENV. There is a period at the end of the definition for LPI-NE but not for any of the other definitions. “U” is defined, but not used anywhere in the table. Please double check for any other typos.

“human health” and “human welfare” seem to be used interchangeably. Sometimes both are listed as “human health and welfare”. Please be clear about the definitions and consistent in how the terms are used.

---

## Round 0.3 · accepted · Accept

After multiple reviews and significant revisions by the authors, I am convinced that this is ready to be published.